# Relation of incident chronic disease with changes in muscle function, mobility, and self-reported health: Results from the Health and Retirement Study

**James Davis**[1]*, **Eunjung Lim**[1], **Deborah A. Taira**[2], **John Chen**[1]

1 John A. Burns School of Medicine, Honolulu, HI, United States of America, 2 Daniel K. Inouye College of Pharmacy, Hilo, HI, United States of America

* jamesdav@hawaii.edu

**Data Availability Statement:** Data are publicly available with registration from the Health and

## Abstract

The primary objective was to learn the extent that muscle function, mobility, and self-reported health decline following incident diabetes, stroke, lung problem, and heart problems. A secondary objective was to measure subsequent recovery following the incident events. A longitudinal panel study of the natural history of four major chronic diseases using the Health and Retirement Study, a nationally representative sample of adults over age 50 years. People first interviewed from 1998–2004 were followed across five biannual exams. The study included 5,665 participants who reported not having diabetes, stroke, lung problems, and heart problems at their baseline interview. Their mean age was 57.3 years (SD = 6.0). They were followed for an average of 4.3 biannual interviews. Declines and subsequent recovery in self-reported health, muscle function, and mobility were examined graphically and modeled using negative binomial regression. The study also measured the incidence rates and prevalence of single and multiple chronic diseases across the follow-up years.Self-reported health and muscle function declined significantly following incident stroke, heart problems, lung problems, and multiple chronic diseases. Mobility declined significantly except following incident diabetes. Self-reported health improved following incident multiple chronic conditions, but recovery was limited compared to initial decline. Population prevalence after five follow-up waves reached 9.0% for diabetes, 8.1% for heart problems, 3.4% for lung disease, 2.1% for stroke, and 5.2% for multiple chronic diseases. Significant declines in self-reported health, muscle function, and mobility occurred within two years of chronic disease incidence with only limited subsequent recovery. Incurring a second chronic disease further increased the declines. Early intervention following incident chronic disease seems warranted to prevent declines in strength, mobility, and perceptions of health.

Retirement web site (https://hrs.isr.umich.edu/data-products).

**Funding:** This study was partially supported by U54MD007601 (Ola HAWAII) from the National Institutes of Minority Health and Health Disparities. The funders had no role in study design, data collection and analysis, decision to publish, or preparation of the manuscript. The HRS (Health and Retirement Study) is sponsored by the National Institute on Aging (grant number NIA U01AG009740) and is conducted by the University of Michigan.

**Competing interests:** The authors have declared that no competing interests exist.

## Introduction

Chronic diseases can lead to declines in physical health and a reduced perception of health. Patients who develop multiple chronic diseases face further risks of adverse outcomes that limit healthy aging [1, 2]. Chronic diseases may decrease function beyond their immediate adverse health effects. For example, studies report that having chronic obstructive pulmonary disease leads to impairments beyond the respiratory system [3–5]. Stroke can affect being able to walk without assistance and safe mobility in the home [6]. A Finland study reported cardiovascular disease and diabetes were long-term predictors of declines in muscle function and a heightened risk of mortality [7]. A study from Taiwan described a gradient effect of function with mortality [8]. Patients with declines in function coupled with multiple chronic diseases had the greatest mortality [8]. A mixed methods study of patients with multiple chronic diseases reported that three of the most bothersome chronic conditions were diabetes, heart failure, and lung problems based on their impact on function and quality of life from symptoms and activity limitations [9]. A British study reported chronic conditions and functional limitations were associated with being pushed from employment, rather than choosing to retire in good health [10].

Having existing chronic conditions can exacerbate the effects of incurring a chronic disease [11]. Patients with multiple chronic conditions often have reduced physical performance, and multiple diseases may increase declines in function with age [12]. One study described patients with congestive heart failure, diabetes, and respiratory problems as having worse physical component scores on the SF-36 over 4 years of follow-up, as did patients reporting four or more chronic conditions [13]. Physical component scores are based on eight scales of SF-36 and higher scores in physical component score indicates better physical health. A cross-sectional study reported patients with 4 or more chronic diseases had poorer physical function, increased disability and a lowered quality of life [14].

We know little about the natural history of health perceptions and function occur relative to incident chronic disease. Our study follows patients even before they incur diabetes, heart problems, lung problems, and stroke using data from the Health and Retirement study (HRS). The objective of this study was to learn how rapidly self-reported health, muscle function, and mobility decline after incident disease and to monitor if health and function recover. We hypothesized that participants would incur declines in function and self-reported health following incident chronic disease and that prior function may not be completely regained.

## Methods

### Data source

The data came from the HRS, a long-term study of aging among a representative national sample of adults in the United States [15]. The HRS provides detailed economic and health data from a series of cohorts designed to cover aging since 1924 to the current time. Cohorts are re-interviewed every two years, which we termed "waves." The HRS provides deidentified data that is freely available with registration from the HRS website. Because the data are deidentified, studies using the HRS data do not require Institutional Review Board approval.

### Study design

We designed the study as a longitudinal analysis of the influence of incident chronic diseases on self-reported health, muscle function, and mobility. The design focuses on incident disease and excluded patients diagnosed with diabetes, heart problems, stroke, or lung problems at their baseline interview. Participants were followed from their baseline exam until the end of

their participation in the study. Participants were not dropped if they missed an interview. Some participants skipped interviews but returned at later exams. We classified participants as having the first disease that occurred if one developed and reclassified them as having multiple chronic conditions if they developed a second of the four chronic diseases.

## Study variables

The questions on chronic diseases were worded "has a doctor ever told you that you have . . .." The conditions were assessed on later interviews to verify the incident disease. Disease would have occurred between the last negative interview and the next interview when the disease was reported. The HRS defined diabetes as having diabetes or high blood sugar; a heart problem as having a heart attack, coronary heart disease, angina, or congestive heart failure; a lung problem as chronic lung disease such as chronic bronchitis or emphysema but except asthma; and stroke as stroke or transient ischemic attack. Having multiple chronic conditions is considered as having at least two of the four chronic diseases—diabetes, heart problems, stroke, and lung problems.

The primary study outcomes were changes in self-reported health, mobility, and muscle function by time since incidence. Secondary outcomes were the incidence and prevalence of diabetes, heart problems, stroke lung problems, and multiple chronic conditions across five follow-up waves. The HRS chose measures suitable for participants with both high and low functional status. Subjective perceptions of health were considered important, regardless of whether they are accurate reflections of objective indicators [16]. The functional status measures were considered to summarize overall health status. Self-reported health had categories of excellent (1), very good (2), good (3), fair (4), and poor (5). These categories were used for the regression model for self-reported health. Self-reported health has been shown to be associated with mortality [17–20]. Mobility was based on being unable to do the following five tasks: walking several blocks, walking one block, walking across the room, climbing several flights of stairs and climbing one flight of stairs. Large muscle function was based on being unable to do the following four tasks: walking one block, walking across the room, climbing one flight of stairs, and bathing. The total number of tasks for mobility (0–5) and large muscle function (0–4) were computed as outcomes. As example, a person able to do all the tasks for mobility or muscle function would get scores of zero. These total scores in regression models for mobility and large muscle function. As example, someone unable to do three of the mobility tasks would get a score of three, and someone unable to do two of the mobility tasks would get a score of 2. A person able to do all of the tasks for mobility or muscle function would get scores of zero. For self-reported health, scores were the category selected (e.g. poor health was scored as five).

## Study participants

The study sample was the participants first enrolled between 1998 and 2004 who, at their first interview, did not report having ever had diabetes, heart problems, lung problems or stroke. We selected these early years so that participants would have multiple follow-up exams. Interviews were conducted in 2004, 2006, 2008, 2010, 2012, and 2014. The study included members of the Children of Depression (CODA) cohort born 1924 to 1930 (n = 1432); members of the War Baby (WB) cohort born 1942 to 1947 (n = 1773), and members of the Early Baby Boomer (EBB) cohort born 1948 to 1953 (n = 2086). Smaller numbers were from the Health and Retirement cohort born 1931 to 1941 (n = 305) and the AHEAD cohort (n = 69) born before 1924. The years of recruitment of early cohorts overlapped by calendar time to provide a wide age range [21]. The cohorts provided a range of participant ages. Response rates tended to be the

lowest in the first interview of the cohort and increase and stabilize thereafter [21]. The initial response rates and subsequent ranges of response rates are 80.4% (87.7%-93.0%) for AHEAD, 81.6% (85.4%-89.6%) for the Health and Retirement cohort, 75.3%-85.5%-87.7%) for EBB, 69.9% (87.0%-90.9%) for WB, and 72.5% (88.7%-92.3%) for CODA. Participants were kept in analyses for all interviews they attended prior to death. Participants were not dropped from analyses if they missed an interview. Some participants skipped interviews but returned at later exams, so they are not lost to follow-up.

## Statistical analysis

We analyzed data in three parts. First, we analyzed the baseline data on demographic variables and functional limitations. Second, we measured incidence and prevalence using data available from the five biannual waves. Third, we used multiple regression to study associations between chronic disease incidence and changes in self-reported health, muscle function, and mobility. Age groups (under age 55, 55–64, 65–74, and 75 years and older), sex, and body mass index (normal body mass, overweight, and obese) were adjusted in all regression analyses.

For the first and second parts, descriptive analyses included means and 95% confidence intervals (CIs) for continuous variables and proportions with 95% CIs for categorical variables. For the third part, we visualized the means of the primary outcomes graphically to understand the patterns of change in function. Changes in function were subsequently estimated as two linear slopes: one prior to incident disease and the other across subsequent waves. The statistical method used was piecewise linear models adjusted for age group, sex and body mass group [22]. Since the outcome variables are provided as counts (i.e., self-reported health score and difficulties with tasks for muscle function and mobility), negative binomial models were utilized controlling for age group, sex, and body mass index. The chronic conditions were those collected by the HRS and not included as outcomes (arthritis, cancer, and psychological problems. Arthritis has been found strongly associated with changes in physical function in older adults [23], cancer and cancer treatments may have an association [24], and psychological problems may have a bidirectional relationship [25].

Results for the change from before to after the incident wave are presented as percentage increases from baseline scores with 95% confidence intervals (CIs) and results for the change across the subsequent follow-up waves are presented as percentage decreases per wave after the incident wave with 95% CIs. All the analyses accounted for the complex survey design of the HRS using the baseline weight and the strata and primary sampling units of the study. All regression models were conducted using generalized estimating equations [26]. We performed analyses using SAS version 9.4 (SAS Institute, Inc., Cary, NC) and R version 4.0.2 (R Foundation for Statistical Computing, Vienna, Austria).

## Results

At their baseline exam the participants had a weighted mean age of 57.2 ± 6.0 years and weighted percentages of 48.15% for female sex, 32.2% for normal weight, 40.0% for overweight, and 27.8% for obesity. Of the three study outcomes weighted means were 2.66 ± 1.13, 0.96 ± 1.21, and 0.71 ± 1.16 for self-reported health, large muscle function, and mobility, respectively. Maximum scores were 5 for self-reported health and mobility, and 4 for muscle function.

Table 1 presents incidence and prevalence for the single and multiple chronic diseases across the follow-up waves. Incidence was less than two percent per year for each of the four of chronic diseases. Incidence only increased every interview for multiple chronic conditions reaching 1.9% at wave 6. Prevalence in contrast to incidence increased every interview for all

**Table 1. Incidence and prevalence of diabetes, heart disease, stroke, lung disease, and multiple chronic conditions by follow-up wave.**

| Condition | Category | Wave 2 | Wave 3 | Wave 4 | Wave 5 | Wave 6 |
|---|---|---|---|---|---|---|
| **Diabetes** | | | | | | |
| | Unweighted N | 102 | 192 | 256 | 333 | 379 |
| | Weighted incidence | 1.90% | 1.70% | 1.60% | 1.50% | 1.40% |
| | Weighted prevalence | 1.90% | 3.80% | 5.40% | 7.70% | 9.00% |
| Heart | | | | | | |
| | Unweighted N | 110 | 193 | 275 | 338 | 371 |
| | Weighted incidence | 1.80% | 1.40% | 1.20% | 1.10% | 1.00% |
| | Weighted prevalence | 1.80% | 3.30% | 5.20% | 7.00% | 8.10% |
| Stroke | | | | | | |
| | Unweighted N | 19 | 47 | 74 | 89 | 105 |
| | Weighted incidence | 0.30% | 0.20% | 0.20% | 0.10% | 0.10% |
| | Weighted prevalence | 0.30% | 0.90% | 1.40% | 1.70% | 2.10% |
| Lung | | | | | | |
| | Unweighted N | 56 | 94 | 123 | 136 | 193 |
| | Weighted incidence | 1.00% | 0.70% | 0.60% | 0.50% | 0.50% |
| | Weighted prevalence | 1.00% | 1.80% | 2.60% | 3.00% | 3.40% |
| Multiple chronic conditions | | | | | | |
| | Unweighted N | 19 | 72 | 121 | 184 | 241 |
| | Weighted incidence | 0.30% | 1.00% | 1.10% | 1.70% | 1.90% |
| | Weighted prevalence | 0.30% | 1.30% | 2.20% | 3.70% | 5.20% |

The total sample size was 5665 at baseline (wave1). Waves are follow-up intervals roughly two years apart. The number of participants at waves 2–6 were 5055, 4870, 4643, 4390, and 4190. Multiple chronic conditions represent having at least two of the four chronic diseases (diabetes, heart problems, stroke, and lung problems).

four chronic diseases and for multiple chronic conditions. At wave 6 diabetes and heart problems attained prevalences of 9.0% and 8.1%, respectively. The prevalence of stroke, lung disease, and multiple chronic conditions reached 2.1%, 3.4%, and 5.2% at wave 6, respectively. Eighty four percent included in the initial wave who had not reported one of the four conditions were know alive at the last follow-up wave.

Fig 1 depicts the means (95% CIs) by chronic condition and wave for self-reported health, large muscle function, and mobility. Higher means correspond to poorer self-reported health, reduced muscle function, and decreased mobility. Across the study conditions, self-reported health has a notable "jump" in means from before-incidence to the first follow-up wave; thereafter, means remained constant or had slight declines. For muscle function and mobility, stroke, lung conditions and multiple chronic conditions had greater changes than diabetes and heart problems (shifts toward less strength and poorer mobility). Following the initial changes muscle function and mobility was not regained. Heart conditions and stroke showed modest improvements in self-reported health.

Results from the adjusted regression models had two coefficients: one for change from baseline to the first follow-up visit and the other to estimate linear slopes across the subsequent waves. Except for diabetes and mobility, the initial coefficients from the piece-wise models showed significantly higher scores representing poorer self-reported health and more difficulties related to large muscle function, and mobility (Table 2). The higher scores for the three outcome measures ranged from 10.6% to 20.5% for self-reported health, from 12.6% to 41.5% for large muscle function, and from 10.7% to 101% for mobility. Participants incurring multiple chronic conditions increased scores by 20.5% for self-reported health, 35.9% for muscle

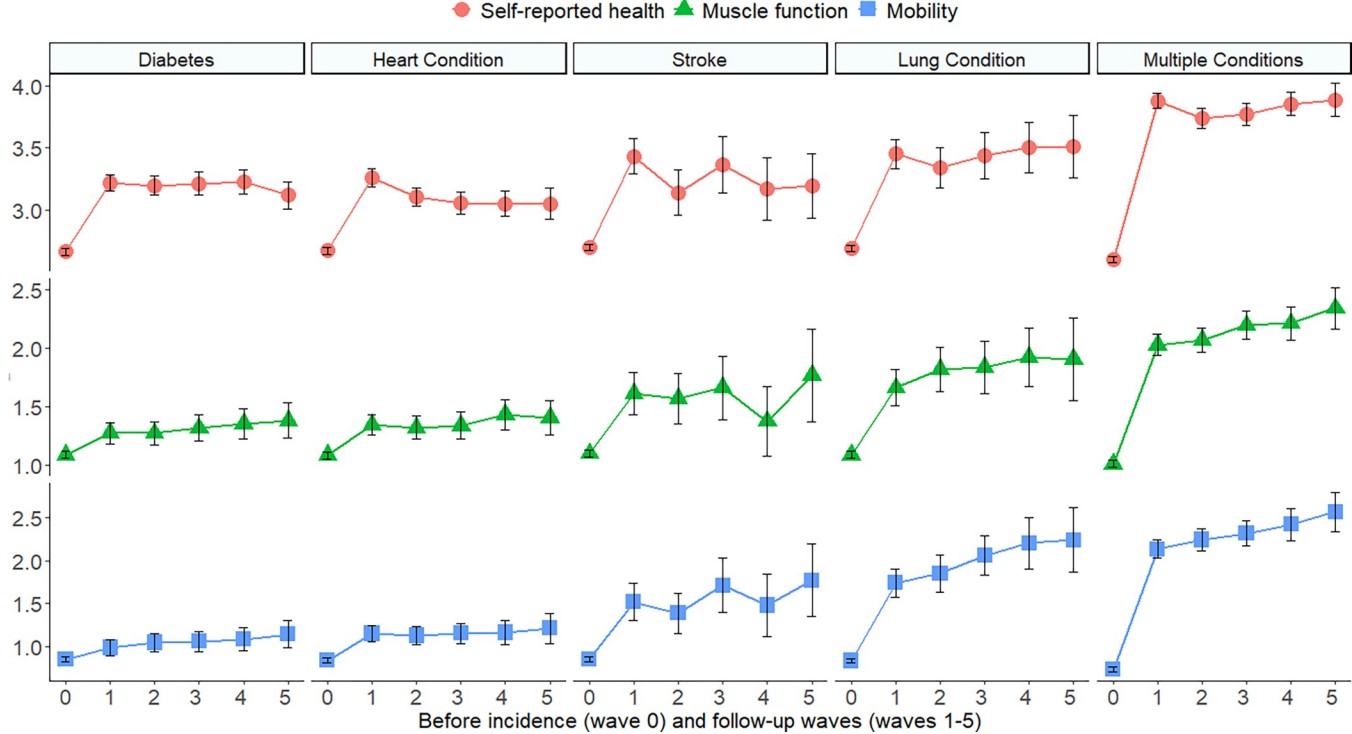

**Fig 1. Means of self-reported health, large muscle function, and mobility by chronic diseases.** Error bars are 95% confidence intervals obtained from piecewise linear regression models. Waves are follow-up intervals roughly two years apart. Patients initially free of the listed chronic conditions were followed from before incidence though five waves after incidence. Patients were analyzed as having single conditions until a second occurred. Subsequently, the patients were treated as having multiple chronic conditions.

function, and 48.1% for mobility. Only mobility for participants incurring multiple diseases showed significant improvement in scores after incident disease occurred (Table 3).

Table 4 gives the associations from the adjusted regression models of age group, sex, and body mass index group with self-reported health, large muscle function and mobility by

**Table 2. Percentage increase from baseline scores at the first wave after disease incidence in self-reported health, large muscle function, and mobility by chronic disease.**

| Condition | Self-reported health | | Muscle function | | Mobility | |
|---|---|---|---|---|---|---|
| | Percentage (95% CI) | P value | Percentage 95% CI | P value | Percentage 95% CI | P value |
| Diabetes | 9.03 (6.29, 11.8) | < 0.001 | 7.31 (-.50, 15.7) | 0.067 | 3.42 (-6.8, 14.8) | 0.528 |
| Heart problems | 14.6 (11.6, 17.6) | < 0.001 | 16.0 (7.07, 25.8) | < 0.001 | 37.0 (22.2, 53.6) | < 0.001 |
| Stroke | 17.6 (11.4, 24.1) | < 0.001 | 26.1 (8.41, 46.6) | 0.003 | 37.7 (11.4, 70.1) | 0.003 |
| Lung problems | 12.8 (7.81, 18.1) | < 0.001 | 13.5 (0.95, 27.5) | 0.034 | 39.8 (15.7, 68.9) | 0.001 |
| Multiple chronic conditions | 15.6 (11.6, 19.8) | < 0.001 | 28.5 (16.3, 42.0) | < 0.001 | 51.2 (29.0, 77.3) | < 0.001 |

Higher scores represent poorer health and declines function. Percentages are the percentage increases at the first wave after disease incidence and CIs are the confidence intervals. Multiple chronic conditions represent having at least two of diabetes, heart problems, stroke, and lung problems. Results are from regression models adjusted for age group (age less than 55, 55–64, 65–74, and 75 years and older), sex, body mass index (normal, overweight, or obese), and self-reported arthritis, cancer, and psychological problems. Waves are follow-up intervals roughly two years apart and participants were followed for up to five waves.

**Table 3. Percentage decreases in scores of self-reported health, large muscle function, and mobility by chronic diseases per wave after disease incidence.**

| Condition | Self-reported health | | Muscle function | | Mobility | |
|---|---|---|---|---|---|---|
| | Percentage (95% CI) | P value | Percentage (95% CI) | P value | Percentage (95% CI) | P value |
| Diabetes | 0.21 (-1.10, 1.50) | 0.754 | 3.77 (-0.58, 7.93) | 0.089 | 1.88 (-4.82, 9.04) | 0.592 |
| Heart problems | 1.38 (-0.09, 2.82) | 0.065 | 2.54 (-1.81, 6.70) | 0.249 | 1.15 (-3.83, 5.89) | 0.645 |
| Stroke | 2.80 (-0.37, 5.87) | 0.083 | 0.45 (-10.3, 10.13) | 0.931 | 4.43 (-7.40, 17.78) | 0.479 |
| Lung problems | 0.77 (-1.64, 3.12) | 0.528 | 1.23 (-4.34, 6.51) | 0.658 | 1.72 (-5.80, 9.84) | 0.664 |
| Multiple chronic conditions | 0.40 (-1.48, 2.24) | 0.676 | 2.18 (-5.15, 8.99) | 0.550 | 6.87 (-2.91, 17.63) | 0.175 |

Lower scores represent better health and improved function. Percentages are the percentage decreases at the first wave after disease incidence and CIs are the confidence intervals. Multiple chronic conditions represent having at least two of diabetes, heart problems, stroke, and lung problems. Results are from regression models adjusted for age group (age less than 55, 55–64, 65–74, and 75 years and older), sex, body mass index (normal, overweight, or obese), and self-reported arthritis, cancer, and psychological problems. Waves are follow-up intervals roughly two years apart and participants were followed for up to five waves.

diabetes, heart problems, lung problems, and stroke. Separate results are given for each of the five disease categories in columns labeled by the diseases. As an example, for diabetes participants age 75 years and older were estimated to have 0.26 higher scores on self-reported health compared to participants under age 55 years. A higher score represents poorer health. For diabetes, large muscle function was estimated as 0.77 units higher for participants age 75 years and older compared to participants under and 55 years, and mobility scores were estimated as 1.94 units higher. Table 4 provides estimates with 95% confidence intervals. Across the five

**Table 4. Relative differences in scores of self-reported health, large muscle function, and mobility by age, sex, and body mass index by single and multiple chronic conditions for diabetes, heart problems, lung problems, and stroke.**

| Outcomes | Predictors | Regression results by Major Chronic Disease | | | | |
|---|---|---|---|---|---|---|
| | | Diabetes | Heart problems | Lung problems | Stroke | Multiple chronic conditions |
| Self-reported health | age 55–59 | 0.07 (0.06, 0.09) | 0.07 (0.06, 0.09) | 0.08 (0.06, 0.09) | 0.08 (0.07, 0.09) | 0.08 (0.06, 0.09) |
| | age 65–74 | 0.13 (0.11, 0.15) | 0.13 (0.11, 0.15) | 0.14 (0.12, 0.16) | 0.14 (0.12, 0.16) | 0.13 (0.11, 0.15) |
| | age 75 and older | 0.26 (0.23, 0.29) | 0.25 (0.22, 0.27) | 0.26 (0.23, 0.29) | 0.26 (0.24, 0.29) | 0.25 (0.22, 0.27) |
| | Female | 0.03 (0.01, 0.05) | 0.03 (0.01, 0.05) | 0.03 (0.01, 0.05) | 0.03 (0.01, 0.05) | 0.03 (0.01, 0.05) |
| | Overweight | 0.05 (0.02, 0.07) | 0.05 (0.02, 0.07) | 0.05 (0.02, 0.07) | 0.05 (0.02, 0.08) | 0.05 (0.02, 0.07) |
| | Obese | 0.17 (0.14, 0.20) | 0.18 (0.15, 0.21) | 0.18 (0.15, 0.21) | 0.18 (0.15, 0.21) | 0.18 (0.15, 0.21) |
| Large muscle function | age 55–59 | 0.24 (0.20, 0.29) | 0.24 (0.19, 0.29) | 0.24 (0.20, 0.29) | 0.25 (0.20, 0.30) | 0.24 (0.19, 0.29) |
| | age 65–74 | 0.35 (0.26, 0.44) | 0.34 (0.25, 0.43) | 0.35 (0.26, 0.44) | 0.35 (0.27, 0.45) | 0.33 (0.24, 0.42) |
| | age 75 and older | 0.77 (0.65, 0.89) | 0.73 (0.62, 0.85) | 0.76 (0.65, 0.88) | 0.76 (0.64, 0.88) | 0.71 (0.60, 0.83) |
| | Female | 0.47 (0.38, 0.57) | 0.47 (0.38, 0.57) | 0.46 (0.37, 0.56) | 0.47 (0.38, 0.57) | 0.47 (0.38, 0.57) |
| | Overweight | 0.17 (0.08, 0.26) | 0.17 (0.08, 0.27) | 0.17 (0.08, 0.27) | 0.17 (0.08, 0.27) | 0.16 (0.07, 0.26) |
| | Obese | 0.62 (0.50, 0.75) | 0.63 (0.50, 0.76) | 0.63 (0.51, 0.77) | 0.63 (0.51, 0.77) | 0.60 (0.48, 0.74) |
| Mobility | age 55–59 | 0.40 (0.32, 0.48) | 0.38 (0.30, 0.46) | 0.38 (0.30, 0.46) | 0.40 (0.32, 0.48) | 0.37 (0.30, 0.45) |
| | age 65–74 | 0.70 (0.55, 0.86) | 0.67 (0.52, 0.83) | 0.68 (0.53, 0.84) | 0.71 (0.56, 0.87) | 0.62 (0.48, 0.77) |
| | age 75 and older | 1.94 (1.68, 2.22) | 1.81 (1.56, 2.09) | 1.87 (1.61, 2.14) | 1.91 (1.66, 2.20) | 1.73 (1.48, 1.99) |
| | Female | 0.65 (0.51, 0.81) | 0.66 (0.51, 0.82) | 0.64 (0.50, 0.80) | 0.65 (0.51, 0.82) | 0.66 (0.52, 0.82) |
| | Overweight | 0.22 (0.09, 0.37) | 0.22 (0.09, 0.37) | 0.23 (0.10, 0.38) | 1.12 (0.89, 1.38) | 0.21 (0.08, 0.36) |
| | Obese | 1.09 (0.86, 1.34) | 1.10 (0.88, 1.35) | 1.11 (0.89, 1.36) | 0.24 (0.10, 0.39) | 1.03 (0.82, 1.28) |

Reference categories are for age groups, being under age 55 years, for sex, being male, and for body mass, having a normal body mass index. Separate regression models for self-reported health, large muscle function, and mobility were adjusted for the rates of declines and recovery across the study waves.

disease categories older age, female, and higher body mass index were associated with higher scores.

## Discussion

The results support the study hypothesis that declines in muscle function, mobility, and self-reported health occur following incident chronic disease and that prior function is not completely regained. We found that self-reported health and large muscle function declined by the next interview, most often held within about two years following incident diabetes, stroke, or heart or lung problems. When graphed, mean scores showed jumps toward worse function for all four study diseases, followed by a leveling off. Adjusted regression models supported that strength, mobility and self-reported health decline after incident disease. We were able to calculate with our study design both the incidence and prevalence of four major chronic diseases across time for older adult population without the diseases at baseline. Incidence increased 1% to 2% per wave for diabetes and heart problems and less than 1% per wave for stroke and lung problems. After five biannual follow-up waves 2.1% of participants had developed stroke, 3.4% had lung problems, 8.1% had heart problems, and 9.0% of participants had developed diabetes. These results show the health risks facing initially healthy older adults.

In our study, incident chronic disease was strongly associated with self-reported health which has strong validation as a measure of adverse health [17–20]. Self-reported overall health is a widely used patient-reported outcome measure for population health surveillance. The strength of an overall self-reported measure is that it conveys not only the objective presence of disease but the patient's perception of the impact of the disease on their overall health. Low self-reported health can predict major cardiovascular events [27]; and people reporting poor health face a greater risk of mortality compared to people reporting excellent health [18–20]. Hence, it may be important for health care providers to monitor self-reported health following chronic disease onset.

Our results found that both perceived health and functional decline follow chronic disease. The combination may have amplified the adverse effects [1]. Effects were not uniform across chronic diseases; mobility declined with incident heart and lung problems and following stroke, but not after incident diabetes. This result is consistent with a study by Fishman using propensity matching that found a weaker association between diabetes and mobility than earlier studies [28]. Fishman assumed that the smaller association was because of closer matching on covariates.

We found only minimal recovery in lost function once declines occurred. A slight rebound occurred in self-reported health following stroke and lung problems, but the rebounds were small compared to the declines. Muscle function and mobility are measures of health status and self-reported health, a risk indicator of mortality. The results emphasize the seriousness of the declines and raise the question of whether immediate interventions might mitigate the effects.

The incidence of having a second chronic condition increased every wave from 0.3% at the first follow-up wave to 1.9% at the fifth follow-up wave. Its prevalence reached 5.2% at the fifth follow-up wave. Having multiple chronic conditions had strong associations with all of the outcomes in our analyses. Earlier research has shown that having multiple chronic conditions affects health outcomes greater than expected from the risks of individual diseases [11, 13, 14]. Multiple chronic conditions can include factors affecting declines beyond multiple chronic diseases. Wei et al. validated an index in the HRS based on 16 measures of function; the index was associated with both physical function and cognitive performance [29]. Given the complexity of patients with multiple chronic conditions, a summary index may be useful for

clinical applications. Early attention to modifiable risk factors may prevent later disability [11–14]. Our results indicate intervention should begin shortly after chronic disease occurs.

In an HRS study Koroukian et al. hypothesized chronic conditions and functional limitations are part of a geriatric syndrome that tilts the balance toward increased patient burden, use of health care services, and costs [1]. Koroukian et al. analyzed eight conditions as measures of the geriatric syndrome (visual impairment, hearing impairment, moderate or severe depressive symptoms, low cognitive performance, persistent dizziness or light-headedness, and severe pain). Outcomes were fair/poor self-reported health and worsening reported health or death within 2-years. The results showed significant increases in prospective health status, major health decline, and mortality with multiple chronic conditions. A commentary by Mohammad on the article noted the importance of learning how much of the decline is preventable and the optimal approaches to prevent the declines [30]. In a follow-up article, Koroukian et al. used a classification tree and regression analysis to find combinations of measures that predicted self-reported health and mortality [27]. They concluded functional limitations and geriatric syndromes were the most prominent.

Social determinants such as isolation and food insecurity can further increase declines in health and functional limitations. Using data from the HRS, Bishop and Wang found food insecurity associated with the number of mobility limitations [31]. Patient self-perceptions of declines may affect their behavior and how they manage their chronic conditions [32–34]. Therefore, further studies are needed to explore the relationships, including mediation and moderation, among social determinants, chronic conditions, and functional limitations.

One strength of the HRS data is that questions on chronic diseases are repeated every interview and discrepancies in reported diseases get resolved. The HRS is a long-standing source of high-quality data on a nationally representative sample of older adults. Our study used a subset of the HRS population free from chronic diseases at baseline so that we could investigate incident disease. However, the following limitations should be noted. First, interviews were two years apart and outcomes were measured at variable times following incidence. Second, the results were controlled for age, sex, and body mass index, but not for health conditions or other social determinants. The results could be biased because of such omitted covariates. Third, the outcomes based on self-reported health and participants more likely to have declines following incident chronic disease may have under or overestimated their muscle function or mobility at their baseline exam leading to differential misclassification. Imprecision in measuring the outcomes by self-report could lead to non-differential misclassification. Forth, in fitting regression lines participant data was limited to interviews in which they participated.

Multiple chronic conditions were a mixed category including patients with different combinations of chronic disease. We did not have sufficient numbers to distinguish combinations of chronic diseases with the greatest adverse effects. Our study based chronic diseases and functional limitations on self-report and may differ from clinical assessments. Functional limitations, however, are based on 4 to 5 questions using validated instruments. Self-reported health is validated against mortality and outcomes of patients with heart disease [18–20]. In our study outcome assessments were at two-year intervals so shorter-term effects could not be analyzed. The change in self-reported health, muscle function, and mobility by the first two-years after incident chronic disease suggests shorter intervals would be useful to study. muscle function.

In conclusion, our study describes the natural history of four major chronic diseases in adults who had reached their fifties without incurring the chronic diseases. This study has significant public health implications. First, our findings suggest that within two years of incident diabetes, stroke, heart problems, or lung problems significant declines occurred in self-reported health, muscle function, and mobility. The only exception was diabetes and mobility.

This suggests that patients may be need additional resources to mitigate or address these deficits, including physical therapy, social support, and ongoing monitoring. For low-income individuals or those with limited health literacy, this support may need to come from community health centers or other community outreach. Approaches to prevent declines should begin when chronic diseases are first identified. Second, we found that multiple chronic conditions heightened the declines. Hence, ambulatory care settings or health plans, including Medicare or Medicaid, may want to flag patients with onset of multiple chronic conditions for more intensive follow-up. Third, the fact that improvements across subsequent years were limited or non-existent emphasizes the importance of preventing the first incident events. Strengthening public health initiatives to lessen modifiable risk factors may prevent later disability.

## Author Contributions

**Conceptualization:** James Davis.

**Formal analysis:** James Davis.

**Methodology:** James Davis, Eunjung Lim, Deborah A. Taira, John Chen.

**Visualization:** James Davis.

**Writing – original draft:** James Davis, Eunjung Lim, Deborah A. Taira, John Chen.

**Writing – review & editing:** James Davis, Eunjung Lim, Deborah A. Taira, John Chen.

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
