## [Decision Letter · Decision Letter 0]

4 Jan 2022

PGPH-D-21-00399

A longitudinal study of declines and subsequent recovery following incident chronic disease of self-reported health, muscle strength, and mobility

Dear Dr. Davis,

Thank you for submitting your manuscript to PLOS Global Public Health. After careful consideration, we feel that it has merit but does not fully meet PLOS Global Public Health’s publication criteria as it currently stands. Therefore, we invite you to submit a revised version of the manuscript that addresses the points raised during the review process.

The manuscript entitled "A longitudinal study of declines and subsequent recovery following incident chronic disease of self-reported health, muscle strength, and mobility" was reviewed by our journal. The reviewers believe there are major concerns that preclude its acceptance in the present form. I invite you to respond to the reviewers' comments and revise your manuscript accordingly. You are required to address the reviewers comments and subject to satisfactory review by the original reviewers will the manuscript be considered for publication. Please be aware that this invitation does not guarantee eventual acceptance of your manuscript.

We look forward to receiving your revised manuscript.

Kind regards,

Geetha Ravindran Menon

Academic Editor

Journal Requirements:

1. Please note that your Data Availability Statement is currently missing the DOI/accession number of each dataset OR a direct link to access each database. If your manuscript is accepted for publication, you will be asked to provide these details on a very short timeline. We therefore suggest that you provide this information now, though we will not hold up the peer review process if you are unable.

2. Please amend your detailed Financial Disclosure statement. This is published with the article, therefore should be completed in full sentences and contain the exact wording you wish to be published.

ii). State the initials, alongside each funding source, of each author to receive each grant.

iii). State what role the funders took in the study. If the funders had no role in your study, please state: “The funders had no role in study design, data collection and analysis, decision to publish, or preparation of the manuscript.”

Additional Editor Comments (if provided):

Reviewers' comments:

Reviewer's Responses to Questions

**Comments to the Author**

1. Does this manuscript meet PLOS Global Public Health’s publication criteria? Is the manuscript technically sound, and do the data support the conclusions? The manuscript must describe methodologically and ethically rigorous research with conclusions that are appropriately drawn based on the data presented.

Reviewer #1: Yes

Reviewer #2: Yes

2. Has the statistical analysis been performed appropriately and rigorously?

Reviewer #1: No

Reviewer #2: I don't know

3. Have the authors made all data underlying the findings in their manuscript fully available (please refer to the Data Availability Statement at the start of the manuscript PDF file)?

Reviewer #1: Yes

Reviewer #2: Yes

4. Is the manuscript presented in an intelligible fashion and written in standard English?

Reviewer #1: Yes

Reviewer #2: Yes

5. Review Comments to the Author

Reviewer #1: A longitudinal study of declines and subsequent recovery following incident chronic disease of self-reported health, muscle strength, and mobility

Introduction

● Explain what physical component score is

● A clear distinction needs to be made between the terms ‘comorbidities’ and ‘multimorbidities’-there is no mention of the term ‘comorbidities’

Methods

● It is unclear what is the study hypothesis is and how did authors test the hypothesis.

● Response rates are very crucial and if the mean of the response rates for all the interviews is provided, it would be useful- it is just stated that the response rate was over 85%

● In the study variables section, while defining the conditions, it has been mentioned as heart diseases while later it has been mentioned as ‘heart problems’ in the same resection which is confusing

● No mention of deaths during the process or loss of follow-up and how was it accounted for

● The authors need to specify how can estimation of self-reported measures of muscle strength and mobility can be valid measures? Why were these measures not verified against gold standard?

● It is possible that several covariates associated with multimorbidity, and muscle strength or mobility can confound the association. For example, glucose levels or mental health status. BMI is an important confounder in the chronic conditions mentioned in the study which hasn’t been adjusted for, neither mentioned as a limitation. The authors need to explain the steps taken by authors to minimize the confounding. Apart from confounding, the study methodology suffers from many sources of selection bias (collider bias/ Berkson’s bias etc) and measurement error (differential misclassification of outcome). The authors need to explain why there is no description of minimizing the sources of systematic error.

Results

● How many years did it take on an average for incident diabetes, stroke, heart problems, and lung problems to lead to decline in self-reported health, muscle strength, and mobility

● Patients developing a second chronic disease experienced additional decline-This is a vague statement-By how much was the decline experienced?

● What is the loss to follow up in the study/in the HRS system?

● To what extent did the combination of ‘other diseases’ like arthritis, cancer and the chronic diseases and multimorbidities influence the mobility and muscle strength in patients as they were considered for the analysis.

● How many patients’ muscle and mobility strength improve with the decline of other diseases like cancer in spite of having diabetes, lung and heart diseases.

● It is unclear why did the authors not present results measures of association and have presented only descriptive statistics and self-reported declines and improvements. The manuscript should be revised to include measures of association.

Discussion

● There is no mention of any bias, and how it was addressed, for eg: a respondent’s bias- as muscle strength and mobility among many others were self-reported.

● Limitations of the study methodology and causal inference are not mentioned in discussion. The authors need to discuss threats to internal validity in the study and how they tried to minimize or control the sources of systematic error.

Reviewer #2: Comments to the authors:

Thank you for considering me to review this manuscript entitled “A longitudinal study of declines and subsequent recovery following incident chronic disease of self-reported health, muscle strength and mobility”. This study aimed to investigate changes in self-reported health, muscle strength and mobility following incident chronic disease. Although the study seems to be relevant from the point of view of public health, I have a few concerns which I have appended below:

Title: Title seems to be a bit confusing. It doesn’t clearly describes the objectives of the study.

Abstract

Design: Duration of longitudinal follow-up is not mentioned.

Page 2, Line 6-9: Time points of measurement not mentioned. Each time point should be clearly mentioned in the manuscript.

Inadequate description of methods.

Page 2, Line 17: I doubt if “multi-morbidity” is the right terminology to use

Page 3, Line 2: The word “physical declines” is inappropriate. Decline in physical health would be appropriate.

Introduction

Line 20: Remove additional comma after follow-up.

Page 4, Line 2: “From before” should be replaced by “even before”

Please state study hypothesis at the end of introduction.

Content of this section appropriate.

Methods

Is there any way by which the sample size of the study could be justified?

Page 4, Line 10: Replace “from before” with “since”

Page 5, Line 8-10: It’s a repetition. Can be removed.

Operationalize heart and lung problems for this study.

Page 6, Line 8: Why skin cancer is kept as an exception?

Line 9: What do you mean by “nervous problems”. It’s not a scientific terminology.

It is strange that no validated questionnaire was used for assessing self-reported health rather a simple likert scale was utilized.

Similarly, no validated test was used for assessing muscle strength. The tasks utilized to assess muscle strength seems to measure mobility and mobility and strength are two different domains.

Statistical analysis

Adjusting age as a covariate in this analysis makes sense. However, covariates such as arthritis, cancer and psychiatric problems are not justified. Why only these diseases were considered covariates and not any other disease?

Couldn’t understand this section (written in a complex manner). Try to simplify.

Discussion

Page 11, Line 19-20: “These results show the risk facing----older adults”- re-phrase

Page 12, Line 14: “Our” is superscripted. Please correct.

Page 13, Line 7: “Limitations”-it’s repeated

Page 8, Line 12-17: There is minimal discussion on main outcome variables of the study.

Page 14, Line 1: Should be “variable times”

Page 15, Line 17: “healthy older adults”-correct

Overall, this section is vaguely written. There is minimal discussion in support of main objective of this study which should be adequately incorporated.

Conclusion

Conclusion of the study is inadequately written. Authors need to strongly state the application of the study on societal level. Clear application to public health should be stated.

6. PLOS authors have the option to publish the peer review history of their article (what does this mean?). If published, this will include your full peer review and any attached files.

**Do you want your identity to be public for this peer review?** For information about this choice, including consent withdrawal, please see our Privacy Policy.

Reviewer #1: **Yes: **Giridhara R Babu

Reviewer #2: No

---

## [Decision Letter · Decision Letter 1]

27 May 2022

PGPH-D-21-00399R1

Relation of incident chronic disease with changes in muscle strength, mobility, and self-reported health: Results from the Health and Retirement Study

Dear Dr. Davis,

Thank you for submitting your manuscript to PLOS Global Public Health. After careful consideration, we feel that it has merit but does not fully meet PLOS Global Public Health’s publication criteria as it currently stands. Therefore, we invite you to submit a revised version of the manuscript that addresses the points raised during the review process.

We look forward to receiving your revised manuscript.

Kind regards,

Geetha R Menon

Academic Editor

Journal Requirements:

Additional Editor Comments (if provided):

Reviewers' comments:

Reviewer's Responses to Questions

**Comments to the Author**

1. If the authors have adequately addressed your comments raised in a previous round of review and you feel that this manuscript is now acceptable for publication, you may indicate that here to bypass the “Comments to the Author” section, enter your conflict of interest statement in the “Confidential to Editor” section, and submit your "Accept" recommendation.

Reviewer #1: All comments have been addressed

Reviewer #2: (No Response)

2. Does this manuscript meet PLOS Global Public Health’s publication criteria? Is the manuscript technically sound, and do the data support the conclusions? The manuscript must describe methodologically and ethically rigorous research with conclusions that are appropriately drawn based on the data presented.

Reviewer #1: Yes

Reviewer #2: (No Response)

3. Has the statistical analysis been performed appropriately and rigorously?

Reviewer #1: Yes

Reviewer #2: (No Response)

4. Have the authors made all data underlying the findings in their manuscript fully available (please refer to the Data Availability Statement at the start of the manuscript PDF file)?

Reviewer #1: Yes

Reviewer #2: (No Response)

5. Is the manuscript presented in an intelligible fashion and written in standard English?

Reviewer #1: Yes

Reviewer #2: (No Response)

6. Review Comments to the Author

Reviewer #1: Thanks for addressing all the comments.

Reviewer #2: Dear Authors,

I am still not satisfied with the author's responses on use if non-validated questionnaires for measuring self-reported health which is an important variable in this study despite existence of several validated tools.

Though muscle strength and mobility are validated against health consitions and are strongly associated but still usage of a tool meant to measure mobility is not justified for strength. Its better not to name the variable as strength.

As authors said that they have replaced the previous covariates with sex and BMI now. Are there any changes in the results with new co-variates? Removing the clinical conditions as co-variates also doesn't solves the matter as they may influence the relationship between dependent and independent variables. Rather the utilization of relevant diseases as covariates shoould be justified. You may keep sex and BMI as coviariates as well.

Discussion of the manuscript can be further improved.

7. PLOS authors have the option to publish the peer review history of their article (what does this mean?). If published, this will include your full peer review and any attached files.

**Do you want your identity to be public for this peer review?** For information about this choice, including consent withdrawal, please see our Privacy Policy.

Reviewer #1: **Yes: **Giridhara R Babu

Reviewer #2: No

---

## [Editor Report · Decision Letter 2]

28 Jul 2022

Relation of incident chronic disease with changes in muscle function, mobility, and self-reported health: Results from the Health and Retirement Study

PGPH-D-21-00399R2

Dear Dr. Davis,

We are pleased to inform you that your manuscript 'Relation of incident chronic disease with changes in muscle function, mobility, and self-reported health: Results from the Health and Retirement Study' has been provisionally accepted for publication in PLOS Global Public Health.

Best regards,

Geetha R Menon

Academic Editor